# Tabersonine Inhibits the Lipopolysaccharide-Induced Neuroinflammatory Response in BV2 Microglia Cells via the NF-*κ*B Signaling Pathway

**DOI:** 10.3390/molecules27217521

**Published:** 2022-11-03

**Authors:** Jiaotai Shi, Chengbo Wang, Chunyan Sang, Stanislas Nsanzamahoro, Tian Chai, Jun Wang, Aimei Yang, Junli Yang

**Affiliations:** 1School of Life Science and Engineering, Lanzhou University of Technology, Lanzhou 730000, China; 2CAS Key Laboratory of Chemistry of Northwestern Plant Resources and Key Laboratory for Natural Medicine of Gansu Province, Lanzhou Institute of Chemical Physics, Chinese Academy of Sciences, Lanzhou 730000, China; 3University of Chinese Academy of Sciences, Beijing 100049, China; 4Shandong Laboratory of Yantai Advanced Materials and Green Manufacturing, Yantai 264000, China

**Keywords:** tabersonine, neuroinflammation, lipopolysaccharide, BV2, NF-*κ*B signaling pathway

## Abstract

The occurrence and development of neurodegenerative diseases is related to a variety of physiological and pathological changes. Neuroinflammation is one of the major factors that induces and aggravates neurodegenerative diseases. The most important manifestation of neuroinflammation is the activation of microglia. Therefore, inhibiting the abnormal activation of microglia is an important way to alleviate the occurrence of neuroinflammatory diseases. In this research, the inhibitory effect of tabersonine (Tab) on neuroinflammation was evaluated by establishing the BV2 neuroinflammation model induced by lipopolysaccharide (LPS). It was found that Tab significantly inhibited the production and expression of nitric oxide (NO), interleukin-1*β* (IL-1*β*), tumor necrosis factor-α (TNF-α), interleukin-6 (IL-6), and reactive oxygen species (ROS) in BV-2 cells stimulated by LPS. In addition, Tab can also inhibit the activation of nuclear factor-*κ*B (NF-*κ*B) induced by LPS, thus regulating inflammatory mediators such as inducible nitric oxide synthase (iNOS). These results indicated that Tab regulated the release of inflammatory mediators such as NO, IL-1*β*, TNF-α, and IL-6 by inhibiting NF-*κ*B signaling pathway, and exerting its anti-neuroinflammatory effect. This is the first report regarding the inhibition on LPS-induced neuroinflammation in BV2 microglia cells of Tab, which indicated the drug development potential of Tab for the treatment of neurodegenerative diseases.

## 1. Introduction

Neuroinflammation is mainly an immune response involving all cells in the central nervous system (including the brain and spinal cord) [1]. The occurrence of neuroinflammation is first reflected in the activation of microglia [2]. Recently, research has found that neuroinflammation is related to the progression of neurodegenerative diseases, such as Alzheimer’s disease (AD), Parkinson’s disease (PD), Amyotrophic lateral sclerosis (ALS), and Huntington’s disease (HD) [3].

Microglia are resident immune cells of the central nervous system [4]. It is usually activated under environmental stress and exposure to lipopolysaccharide (LPS), interferon (IFN)-*γ*, and *β*-amyloid [5,6,7]. After activation, the morphology of microglia exhibits various changes, including cell proliferation, hypertrophy of the cell soma, increased branching, and secretion of a variety of inflammatory factors and neurotoxic substances, such as interleukin-1*β* (IL-1*β*), tumor necrosis factor-α (TNF-α), interleukin-6 (IL-6), nitric oxide (NO), and reactive oxygen species (ROS) [1,8]. Therefore, inhibiting the release of pro-inflammatory mediators caused by abnormal activation of microglia will be a treatment to slow down neurodegenerative diseases [9].

Tabersonine (Tab, Figure 1A), mainly isolated from the medicinal plant *Catharanthus roseus* (L.) G. Don, has a variety of biological activities including anti-inflammatory [10], and anti-cancer [11] effects. It is reported that Tab has potential anti-inflammatory effects and protects against lung injury by inhibiting NF-*κ*B and P38/JNK pathways to significantly reduce the inflammatory response of lung tissue and inhibiting the production of pro-inflammatory mediators by BMDMs cells [12]. Another research study shows its mechanism through cultured cardiomyocyte-like H9C2 cells and primary rat myocardium, indicating that Tab targets to inhibit TAK1-mediated inflammatory cascade reaction and has a protective effect on AngII-mediated myocardial injury [13]. However, it has not been reported whether Tab could improve neuroinflammation. Herein, the present study was designed to determine whether Tab can alleviate neuroinflammation, thereby mitigating the induction of neurodegenerative diseases. It was demonstrated that Tab can inhibit LPS-induced neuroinflammation in BV2 microglia cells by inhibiting the activation of TLR4 and its downstream target NF-*κ*B.

## 2. Result

### 2.1. Effect of Tab on the Viability of BV2 Cells

In order to determine the effect of Tab on the viability of BV2 cells, The cells were treated with different concentrations at 1, 3, 6, 8, 10, and 20 μM for 24 h. The results showed that Tab did not affect the cell viability in the concentration range of 1–10 μM (Figure 1B). In the subsequent experiment, the non-cytotoxic concentration was used (≤10 μM). In addition, Tab combined with LPS treatment was used to determine its effect on the cell viability of BV2. The cells were pretreated with different concentrations of Tab at 1, 3, 6, 8, and 10 μM for 4 h, followed by co-treatment with LPS of 1 μg/mL for 24 h. The results showed that Tab combined with LPS treatment of BV2 did not show cytotoxicity, and there was no significant difference compared with the control group (*p* > 0.05) (Figure 1C). As shown in the above results, Tab treatment at a concentration of 1–10 μM had no cytotoxicity to BV2. In addition, combined LPS treatment also had no cytotoxicity. In order to understand the effect of Tab on the morphology of microglia induced by LPS, the morphological changes in microglia were observed by immunostaining with anti-CD11b (microglial marker) antibody. As shown in Figure 1D, the microglia treated with LPS showed irregular shape and amoeba state. Tab alleviated the morphological changes in microglia induced by LPS, showing round and small cells in resting state.

### 2.2. Effect of Tab on LPS-Induced Neuroinflammation

The anti-inflammatory effect of Tab was evaluated by detecting the release of NO, the main inflammatory mediator produced by LPS-induced BV2 cells [14]. The release of NO from the culture supernatant was determined indirectly by the Griess method. The results showed that LPS alone significantly induced NO production compared with the untreated control group (*p* < 0.001) (Figure 2A). However, Tab significantly reduced LPS-induced NO production in BV2 cells in a concentration-dependent manner ranging from 3 to 10 μM. A western-blot (WB) analysis showed that the production of NO in BV2 cells stimulated by Tab, and LPS was related to the decreased expression of iNOS. As shown in Figure 2C, LPS of 1 μg/mL could significantly up-regulate the expression of iNOS in BV2 cells (*p* < 0.001), while Tab treatment significantly inhibited the expression of iNOS in BV2 cells induced by LPS (*p* < 0.001).

### 2.3. Effect of Tab on LPS-Induced Inflammatory Factors in BV2 Cells

The anti-inflammatory properties of Tab were evaluated according to the secretion of IL-1*β*, TNF-*α*, and IL-6 LPS-induced in BV2 cells. The contents of IL-1*β*, TNF-*α*, and IL-6 in the supernatant of the culture medium were determined by the ELISA method. According to the results of inflammatory cytokines, LPS alone significantly increased the levels of IL-1*β*, TNF-*α*, and IL-6 in BV2 cells, compared with the control group (*p* < 0.001), while Tab treatment significantly inhibited the production of IL-1*β*, TNF-*α*, and IL-6 in cell supernatant (*p* < 0.001) (Figure 3A–C).

### 2.4. Effect of Tab on LPS-Induced ROS in BV2 Cells

In order to detect the effect of Tab on the release of ROS from BV2 cells induced by LPS, the effect of fluorescent probe DCFH-DA on BV2 cells was detected by flow cytometry. The results showed that compared with the control group, the release of ROS was significantly increased after LPS induction, while the level of ROS was significantly decreased by Tab at 3, 6, and 10 µM (*p* < 0.001) (Figure 4).

### 2.5. Tab Inhibits LPS-Induced Microglia Activation through NF-κB Pathway

Finally, the molecular mechanism of Tab of inhibiting the release of NO, iNOS, and inflammatory factors was clarified by BV2 cell based assay. The NF-*κ*B pathway plays an important role in pathological neuroinflammation [1]. In the activated state, NF-*κ*B regulates the expression of pro-inflammatory genes through IKK*β* phosphorylation, I*κ*B*α* degradation, and subsequent p65 translocation to the nucleus [15,16,17]. The results of western blot (WB) assay showed that in BV2 cells activated by LPS, the expression of p-IKK and p-p65 increased significantly, while the expression of I*κ*B*α* decreased significantly. Under the treatment of Tab, the phosphorylation of IKK and p65 was significantly inhibited (Figure 5B,D) and the expression of I*κ*B*α* protein was increased (Figure 5C). The results showed that Tab significantly inhibited the activation of IKK, thereby inhibiting the LPS-induced degradation of I*κ*B*α*, reducing the nuclear translocation of NF-*κ*B p65, and regulating the expression of pro-inflammatory diseases.

## 3. Discussion

Research has shown that natural products and/or natural product structures play an important role in drug discovery and development. Of 1881 drugs developed from 1981 to 2019, 49.2% were directly developed from natural products or natural product derivatives [18]. In this study, we proposed that, for the first time, a natural compound Tab, mainly from *C. roseus* [13], can inhibit the occurrence of neuroinflammation LPS-induced in BV2 cells.

Microglia plays an important role in the regulation of brain microenvironment and are usually found in an inactivated state [19]. Once activated by stimulation, microglia will undergo morphological changes and secrete a variety of inflammatory factors and neurotoxic substances to cope with changes in the surrounding environment [1]. These inflammatory factors and toxic substances further trigger inflammatory cascade reactions, leading to neuronal apoptosis or necrosis, resulting in neurofunctional damage [20]. In this study, after BV2 cells were treated with LPS, the number of cell branches increased, and the cell body enlarged and formed more long processes, while Tab alleviated the change in cell state, decreased cell branches and cell body. In addition, Tab significantly inhibited the increased expression of IL-1*β*, TNF-*α*, and IL-6 and ROS induced by LPS. Moreover, it was reported that under physiological conditions, the concentration of NO was maintained at a low level and participates in normal metabolic activity and cell proliferation as an intracellular messenger [21]. Under pathological conditions [22], the expression of inducible nitric oxide synthase (iNOS) in microglia increased, resulting in the production of its product NO. This study shows that Tab reduced the release of NO induced by LPS by inhibiting the upregulation of iNOS. Tab inhibited the production of these inflammatory mediators, indicating that it plays an important role in relieving neuroinflammation.

TLR4 is widely expressed in a variety of cell membranes of CNS. Stimulated by endotoxin and excessive release of ROS, it can activate the NF-*κ*B signaling pathway of microglia and regulate the expression of inflammatory mediators, such as IL-1*β*, IL-6, TNF-*α*, and iNOS [23,24]. NF-*κ*B signaling pathway is a classic pathway of microglia-mediated inflammation, which is involved in the regulation of many genes in neuroinflammatory response [25]. Normally, NF-*κ*B binds to its inhibitor I*κ*B to form a complex that exists in the cytoplasm in an inactive form. However, when I*κ*B is phosphorylated and degraded, NF-*κ*B is translocated to the nucleus, which induces the transcription of various genes related to inflammation and apoptosis and promotes neuroinflammation [24,26]. In the present study, the results showed that Tab inhibited the activation of IKK, thus inhibiting the degradation of I*κ*Bα induced by endotoxin, reducing the nuclear translocation of NF-*κ*B p65, and regulating the expression of pro-inflammatory genes, and finally alleviating the development of neuroinflammation (Figure 6).

In summary, this study shows that Tab can attenuate the activation of microglia induced by LPS by regulating the NF-*κ*B signal. Further research is necessary to verify the protective effect of Tab on neuroinflammation in vivo.

## 4. Materials and Methods

### 4.1. Cell Culture

The BV2 microglial cells (provided by BeNa Culture Collection) were cultured in high glucose DMEM (Gibco, Carlsbad, CA, USA) supplemented with 10% FBS (BI, Haifa, Israel) and 1% penicillin (100 U/mL)/streptomycin (100 μg/mL) (Beyotime, Shanghai, China) in an atmosphere of 5% CO_2_ at 37 °C [27].

### 4.2. Cell Viability

The effects of Tab and LPS (ACMEC, Shanghai, China) on the viability of BV2 cells were determined by the CCK-8 method. BV2 cells in the logarithmic growth phase were inoculated in 96-well plates, and the number of cells was 6 × 10^4/^mL. After being treated with different concentrations of Tab (1, 3, 6, 8, 10, and 20 μM) for 4 h, LPS (1 μg/mL) was added for 24 h. Then, 10 μL CCK-8 was added to each well for 1 h, and the absorbance was measured at 450 nm using a microplate reader (Rayto, Shenzhen, China) for statistical analysis [14].

### 4.3. Nitric Oxide Assay

The content of NO produced by cells was measured indirectly by the Griess reagent method. BV2 cells in the logarithmic growth phase were pretreated with Tab for 4 h and treated with LPS (1 µg/mL) for an additional 24 h in 48-well plates. The number of cells was 10 × 10^4^/mL. Then, the culture supernatants were collected and mixed with detection reagents (Beyotime, Shanghai, China). The level of sodium nitrite was detected at 562 nm and the concentration of nitrite is determined according to the standard curve of known sodium nitrite concentration [14].

### 4.4. Detection of Inflammatory Factor Content

ELISA was used to measure the release of IL-1*β* (Beyotime, Shanghai, China), TNF-*α* (Beyotime, Shanghai, China), and IL-6 (Beyotime, Shanghai, China) in BV2 cells induced by LPS. In a 48-well plate, the cells (10 × 10^4^/mL) were pretreated with Tab for 4 h, then treated with LPS (1 μg/mL) for 24 h, and the culture supernatant was collected. The supernatant of the cell supernatant was centrifuged for 120× *g* by 10 min and then detected by double antibody sandwich ELISA method. The contents of IL-1*β*, TNF-*α*, and IL-6 were detected according to the manufacturer’s method [14].

### 4.5. Immunocytochemical Staining

After the cell climbing piece is laid in a 24-well plate, the cells (10 × 10^4^/mL) were pretreated with Tab for 4 h, then treated with LPS (1 μg/mL) for 24 h. BV2 cells abandoned the culture medium, washed with PBS three times, fixed for 20 min with 4% paraformaldehyde at room temperature, and then 15 min with 0.3% TritonX-100 permeability. Then the cells were sealed at 37 °C for 1 h with an immunostaining blocking solution. Added anti-CD11b (1:80) (FITC-65055, Proteintech, Wuhan, China) and incubated at 4 °C overnight. On the second day, the cells were incubated with FITC- conjugated secondary antibody (1:100) (Invitrogen, Carlsbad, CA, USA) at 37 °C for 1 h, and washed three times with PBS. After 5 min was incubated with DAPI (BioSharp, Hefei, China) at room temperature, the cells were observed and photographed under fluorescence microscope [27].

### 4.6. Reactive Oxygen Species Assay

BV2 cells in the logarithmic growth phase were pretreated with Tab for 4 h and treated with LPS (1 µg/mL) for an additional 24 h in 6-well plates. The number of cells was 10×10^4/^mL. Then, the culture supernatants were collected and 10 μmol/L DCFH-DA (Beyotime, Shanghai, China) was added to incubate 20 min at 37 °C. The cells were washed with serum-free medium three times and analyzed immediately on flow cytometry (Agilent NovoCyte, Palo Alto, CA, USA) with excitation wavelength of 488 nm and emission wavelength of 525 nm [24].

### 4.7. Western Blot

BV2 cells were treated with Tab (3, 6 and 10 μM) for 4 h, and then treated with LPS (1μg/mL) for 1 h. Then, the cells were lysed by Cell lysis buffer for Western and IP (BioSharp, Hefei, China), added to the sample buffer, cooked, and stored −80 °C. The cell lysate (20 μg protein) was separated by SDS-PAGE (BioSharp, Hefei, China) and electroprinted on polyvinylidene fluoride (PVDF) membrane. The antibodies against p65 (#8242s, Cell Signaling Technology, Danvers, MA, USA), p-p65 (#3033s, Cell Signaling Technology, Danvers, MA, USA), IKK (#61294s, Cell Signaling Technology, Danvers, MA, USA), p-IKK (#2697s, Cell Signaling Technology, Danvers, MA, USA), IκBα (#4812s, Cell Signaling Technology, Danvers, MA, USA), iNOS (#13120s, Cell Signaling Technology, Danvers, MA, USA), and β-actin (TA-09, zsbio, Beijing, China) were sealed with 5% skimmed milk powder for 2 h and incubated overnight at 4 °C. Washed with TBST buffer three times, diluted with anti-rabbit (ZB-2301, zsbio, Beijing, China) or anti-mouse (ZB-2305, zsbio, Beijing, China) IgG horseradish enzyme in TBST (1:8000), and incubated at room temperature for 2 h. After being treated with chemiluminescent agent (BioSharp, Hefei, China), the protein bands were detected by chemiluminescence instrument (SAGECREATION, Beijing, China) and quantified by ImageJ 1.52a software (National Institutes of Health, Bethesda, MD, USA) [14].

### 4.8. Statistical Analyses

Data are presented as the mean ± SEM (*n* = 3). Statistical analyses were performed using the software GraphPad Prism 8.0.2. Data were tested for normality using the Kolmogorov–Smirnov or the Shapiro–Wilk test, and equality of variance was confirmed using the F-test. Data from multiple groups were analyzed by one-way analysis of variance (ANOVA) followed by Tukey’s tests. *p* < 0.05 was considered significant [28,29].

## Figures and Tables

**Figure 1 molecules-27-07521-f001:**
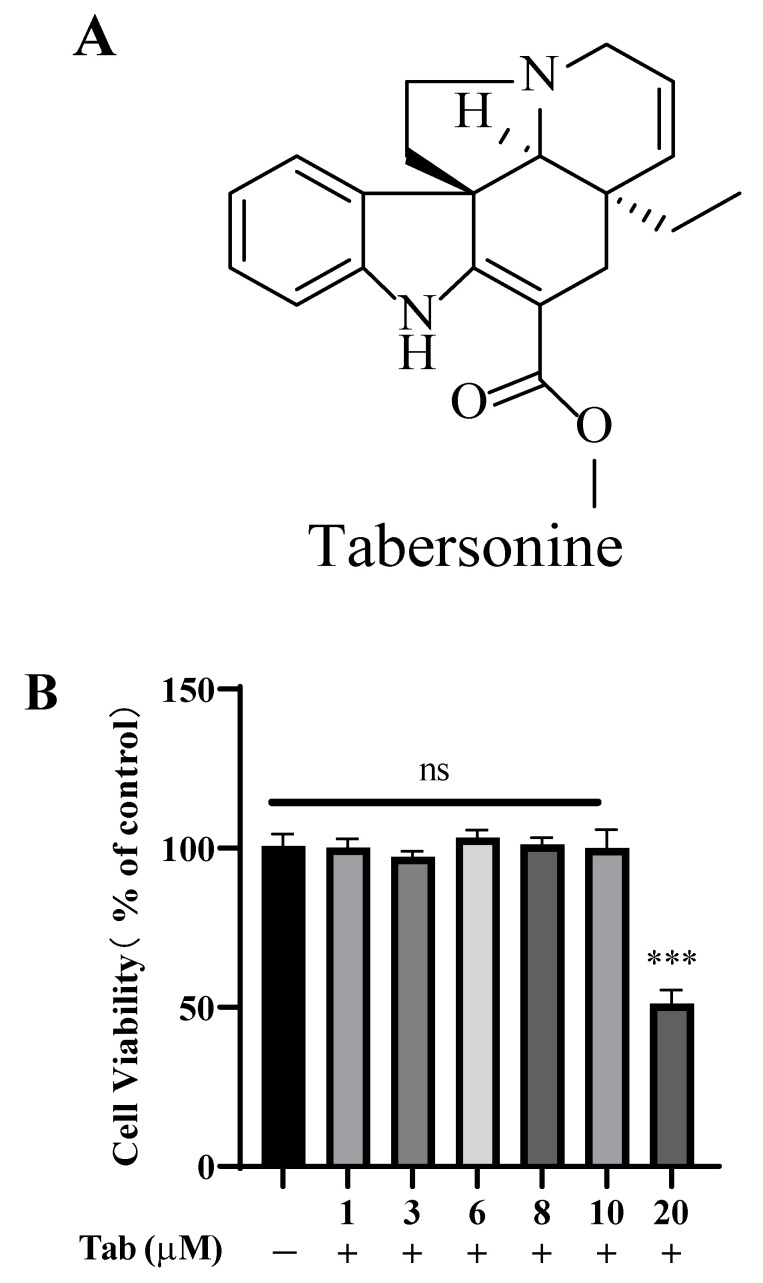
Effect of Tab on the viability of BV2 cells. (**A**) The chemical structure of Table (**B**) BV2 cells were treated with 0, 1, 3, 6, 8, 10, and 20 μM Tab for 24 h. (**C**) The cells were treated with 0, 1, 3, 6, 8, and 10 μM Tab for 4 h, then treated with LPS (1 μg/mL) for 24 h. The cell survival rate was determined by CCK-8. (**D**) Microglial activation was visualized by immunostaining with an anti-CD11b antibody. *** *p* < 0.001 vs. control cells.

**Figure 2 molecules-27-07521-f002:**
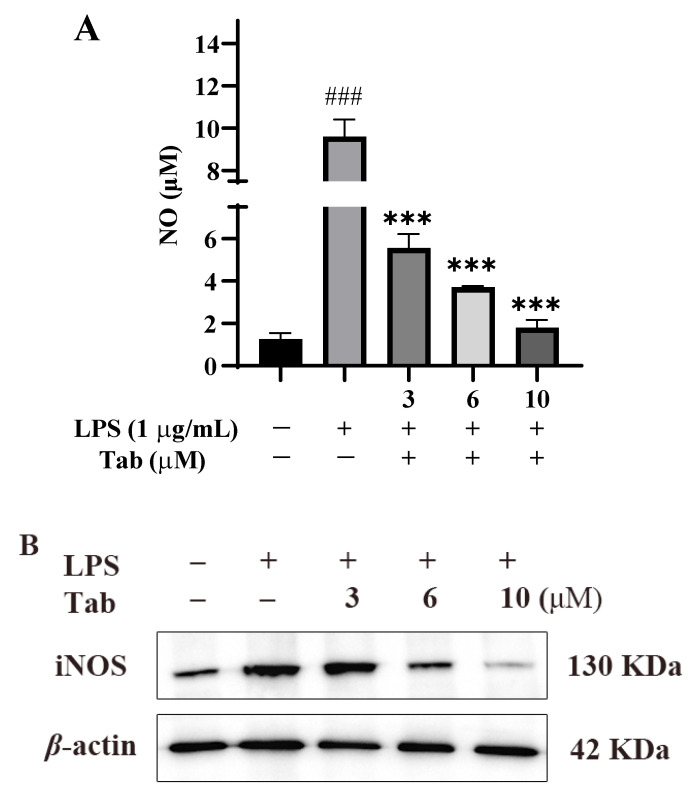
Effect of Tab on neuroinflammation induced by LPS in BV2. After the cells were pretreated with Tab for 4 h, the supernatant was collected after being stimulated by LPS for 24 h. (**A**) The content of NO. (**B**) The protein was extracted. The effect of Tab on the expression of iNOS was detected by Western blot. (**C**) The relative expression of iNOS in (**B**). ### *p* < 0.001 vs. untreated control cells and *** *p* < 0.001 vs. LPS-treated cells.

**Figure 3 molecules-27-07521-f003:**
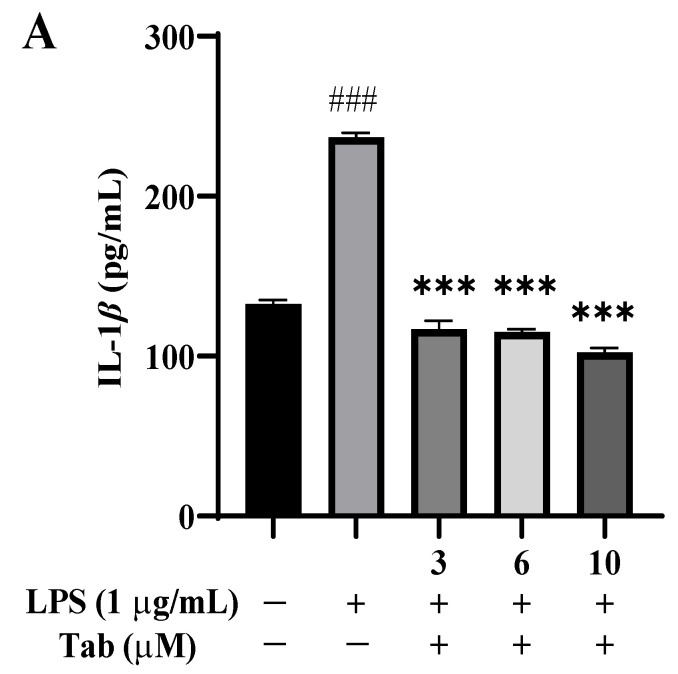
Effect of Tab on the production of IL-1*β*, IL-6, and TNF-*α* in BV2 cells induced by LPS. The cells were pretreated with Tab (3, 6, or 10 μM) for 4 h, then treated with LPS (1 μg/mL) for 24 h. The culture supernatant was collected to detect the production of IL-1*β* (**A**), TNF-*α* (**B**), and IL-6 (**C**). ### *p* < 0.001 vs. untreated control cells and *** *p* < 0.001 vs. LPS-treated cells.

**Figure 4 molecules-27-07521-f004:**
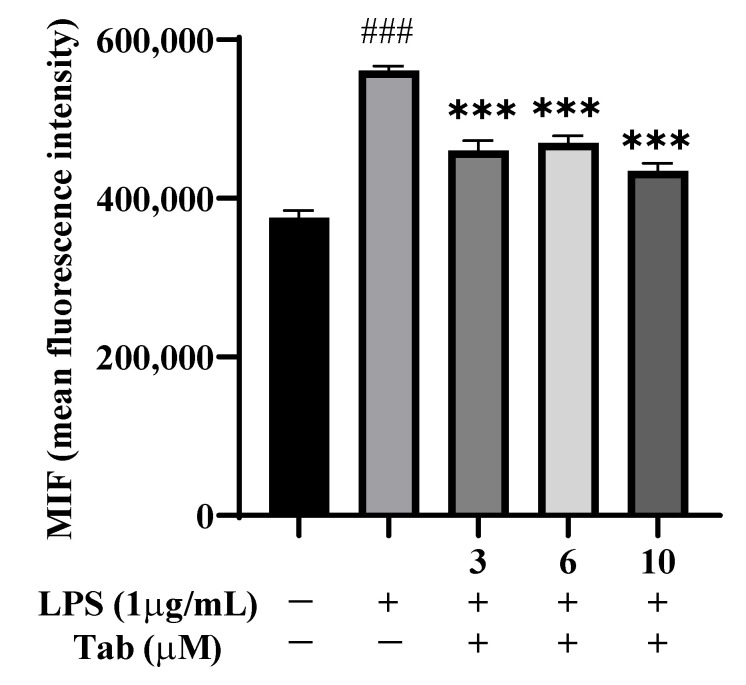
Effect of Tab on LPS-induced ROS in BV2 Cells. The cells were pretreated with Tab for 4 h, then treated with LPS (1 μg/mL) for 24 h. Then quantitative analysis of DCFH-DA accumulation by FACS. ### *p* < 0.001 vs. untreated control cells and *** *p* < 0.001 vs. LPS-treated cells.

**Figure 5 molecules-27-07521-f005:**
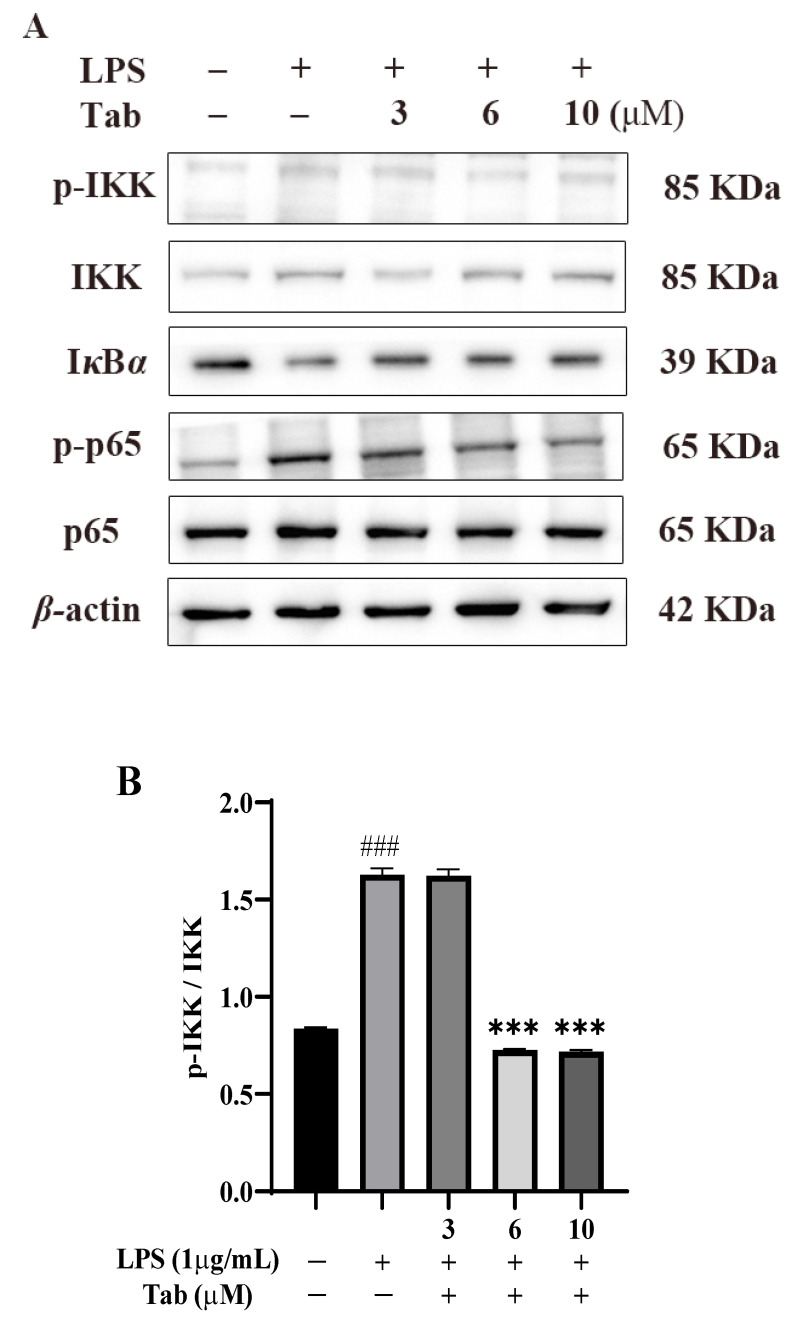
Tab inhibits LPS-induced microglia activation through NF-*κ*B pathways. BV2 cells were treated with Tab (3, 6, and 10 µM) for 4 h followed by the treatment with LPS (1 μg/mL) for 1 h. (**A**) The effect of Tab on the expression of p-p65, p65, I*κ*B*α*, p-IKK, and IKK was detected by Western blot. (**B**–**D**) The relative expression of p-IKK/IKK, I*κ*B*α*, and p-p65/p65 in (**A**). ### *p* < 0.001 vs. untreated control cells; ** *p* < 0.01 and *** *p* < 0.001 vs. LPS-treated cells.

**Figure 6 molecules-27-07521-f006:**
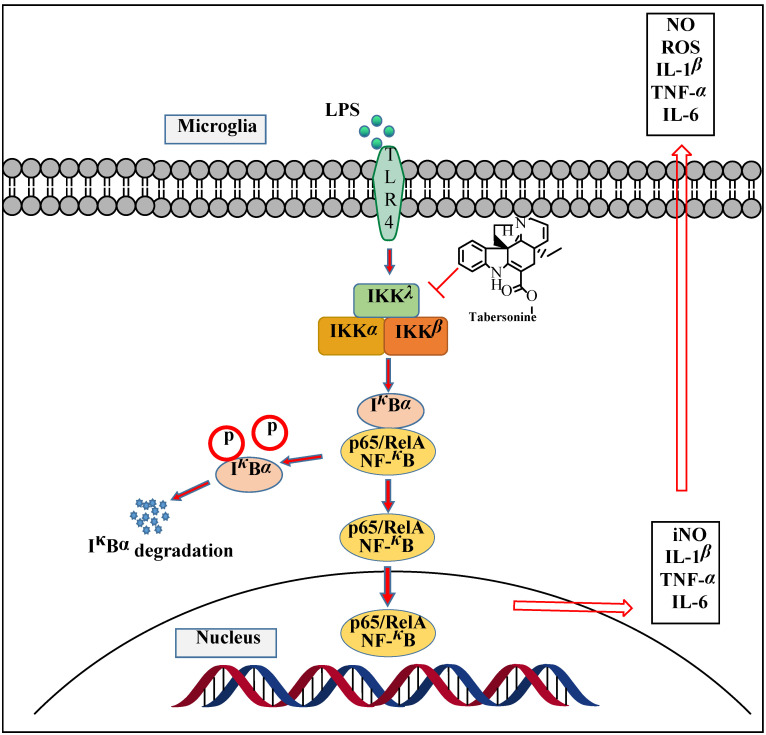
A schematic diagram for the potential molecular mechanism of the anti-neuroinflammatory effect of Tab through the NF-*κ*B pathway.

## Data Availability

Data are contained within the article.

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
