# Peer review of "Tabersonine Inhibits the Lipopolysaccharide-Induced Neuroinflammatory Response in BV2 Microglia Cells via the NF-κB Signaling Pathway"

_molecules, 2022, doi:10.3390/molecules27217521_

Round 1

Reviewer 1 Report

The manuscript is well written but the use of could in the article should be revised to reported speech and authors should endeavour to discuss the results using relevant literature while avoiding excessive review of literature under discussion

Reviewer 2 Report

Dear Author,

Thanks for submitting your research manuscript entitled "Tabersonine inhibits LPS-induced neuroinflammatory response in BV2 microglia cells via NF-κB signaling pathway".

Before giving my final comments as well as the final revision of this manuscript, the author needs to address the following comments scientifically.

Major concerns:

Please find out the following comments:-

·         The rationale and purpose behind selecting the selection of BV2 microglia cells via NF-κB signaling pathway in neuroinflammatory responseis explained very poorly, irrelevant and in incomplete manner throughout the manuscript.

·         Lack of update references with incomplete experimental design is another major concern.  

·         Title, and abstract is misleading the reader. Title needs to reframe in simply manner accordingly.

·         Rationale, Selection and evaluation of several cellular and molecular targets in association with memory defect is very poorly explained, and justified in abstract, intro as well in discussion part.

·         The reviewer found irrational and non-scientific justification in the abstract—introduction and discussion part.

·         Abstract is very poorly written and very confusing. Irrational and fused with repetitions. The reviewer found irrational and non-scientific justification in the abstract—introduction and discussion part.

Example 1: Neuroinflammation is closely related to the occurrence and development of central nervous system diseases.????????????

Example 2: Tab could significantly inhibit the expression of NO, IL-1β, TNF-α, IL-6 and ROS induced by LPS at the concentrations of 3, 6 and 10 μM. In addition, Tab inhibited LPS-induced activation of NF-κB, thereby regulating the release of inflammatory mediators. The results showed that Tab could regulate the release of inflammatory mediators such as NO, IL-1β, TNF-α and IL-6 by inhibiting NF-κB signaling pathway, and exerting its anti-neuroinflammatory effect.?????????

Example 3: This is the first report regarding the inhibition on LPS-induced neuroinflammation in BV2 microglia cells of Tab, which indicated the drug development potential of Tab for the treatment of neurodegenerative diseases.???? What authors want to say? The incomplete justification and scientific correlation is another concern.

·         The results and discussion are very poorly explained. Reviewer surprise to see the justification at the end of discussion part “In order to determine whether NF-κB is involved in the anti-inflammatory effect of Tab, we detected its effects on NF-κB p65 phosphorylation, IKK phosphorylation and IκB in LPS-treated BV2 cells by WB methods. Our results show that Tab inhibits the activation of IKK, thus inhibits the degradation of IκBα induced by endotoxin, reduces the nuclear translocation of NF-κB p65, and regulates the expression of pro-inflammatory genes. In summary, this study shows that Tab attenuates the activation of microglia induced by LPS by regulating the NF-κB signal. Further research is necessary to verify the protective effect of Tab on neuroinflammation in vivo”Author need to directly strike in scientific and readily manner. And simplify whole manuscript directly focus on incidence of actual concern and remove all lines, paragraphs that are saying irrelevant correction with word “First time” etc……..

·         The reviewer feels the author needs to elaborate and justify it with proper citations and strong evidence. The author fails to explain the relevant justification in the introduction as mentioned in the discussion part.

·         A major drawback is a lack of supporting pre-clinical and clinical evidence regarding targeting drugs.

·         Complete mismatch of abstract, introduction, results and discussion in concern with effective smoking cessation agents. Author didn’t justify specific.

Title:

·         Mismatch of title with relevant introduction and conclusive remarks in the conclusion part.

Abstract:

-     The rationale behind this research is not well explained, and several major concerns still constrain the reviewer's enthusiasm for publishing this manuscript.
Introduction:

- The basic literature is not well written and does not even include any literature on alternative approaches with updated references regarding involvement of current drug treatment/techniques used in neuroinflammation.

- Authors fail to justify the correlation, and almost irrational and common information is present in the introduction part.

Material and methods:

-     Major drawback is the lack of supporting references and incomplete experimental and behavioral paradigms.

- Provide biochemicals kits numbers along with their city, country in all individual parameters in all expressions, blots, etc.

- In order to support the assessment of all mentioned parameters in his study, the author should provide all the source documents and data he/she has followed for all assays and estimates.

- How was the dosing determined? Dose-responses should be performed.

- How was the sample size determined? Ideally, a priori sample size calculation should be performed to determine the appropriate sample size.
- Normality and variance homogeneity should be assessed across all groups of the same outcome variable and not individual experimental groups. If the data were not normally distributed or variance homogeneity was not met, nonparametric tests need to be performed.
Parametric data should be reported as mean +/- SD, while nonparametric data should be given/displayed as median and interquartile range. Longitudinal data should be analyzed using repeated measures tests.

Results:

-          All results are very poorly explained. Revised all.

-          All blot analysis, and (Figure 1d) are highly blurred and there is no clarity for easy understanding. Not acceptable in current form.

-          Re-check all figures 1b, 1c, and confirm either statistical symbol are properly mentioned in graphs or not?

-          Page number 5, line number 100, only symbols are there. Recheck????

-          Results need more clarification and significant justification. Differentiating between the outcome and the discussion sections is quite difficult.

-          Must provide all results description and Use proper statistical reporting: i.e. for the results of each statistical test, the authors should report the statistical test that was applied, the test statistic (e.g. t, U, F, r), degrees of freedom as subscripts to the test statistic, and the exact probability value, including those for normality and variance homogeneity tests. Statistics should be reported in APA format, i.e.: t(df) = value, p = value; F(df1,df2) = value, p = value; r(df) = value, p = value; [chi]2 (df, N = value) = value, p = value; Z = value, p = value.  Include statements on the tests for normality and variance heterogeneity and respective results. If the data were not normally distributed or variance heterogeneity was not met, nonparametric tests need to be applied.

Discussion:

-     To address the outcome of in-vivo measures/results separately and how they correlate with the existing literature, it would be better if the author restructured to take a more critical approach for effective neuroinflammation.
-     In the discussion and the conclusion, the aims, rationale, and future perspectives are not evident clearly in relation with in-vitro and in-vivo experimentation.
-     The discussion is usually unorganized at the beginning to address all the observations and evaluate them at the end. It makes the results easier to contextualize and simpler to comprehend.

- Furthermore, a minimal critical analysis should be provided, along with current study limitations as well the future perspective as separate paragraph.

Conclusion:

-          Need to revise the conclusion in a scientific manner. Not accepted in its current form.

-          This reviewer considers that this paper cannot be published in the present form. A detailed revision shortening, ordering and following the commented ideas could improve this interesting paper in a significant manner.

-          Several typewriting mistakes are present and needing correction. This reviewer remains at entire disposal for the next version.

Reviewer 3 Report

The authors provide the results of Tab treatment of lipopolysaccharide (LPS) induced inflammation of BV2 microglia cells. They report that Tab exerted an anti-inflammatory effect through inhibition of the NF-κB signaling pathway.

Title:

I suggest a small change to the title:

Tabersonine inhibits the lipopolysaccharide-induced neuroinflammatory response in BV2 microglia cells via the NF-κB signaling pathway

Abstract:

The authors should use “lipopolysaccharide (LPS)” the first time LPS is mentioned.

Introduction:

Overall the introduction is appropriate and I suggest the following editorial changes:

Please change “…vitro model for the studying of neuroinflammation [2,7-8]” to …vitro model for the study of neuroinflammation [2,7-8].

Please change “Another research shows that Tab…” to Another research study shows that Tab…

In the paragraph starting with “Tabersonine (Tab, Figure 1A), mainly isolated from the medicinal plant Catharanthus…” please indicate which cell lines were used in the referenced studies.

In the last sentence of the introduction, please change “It was found that…” to In this study, it was demonstrate that…

In the last sentence of the introduction, please change “of TLR4 and its downstream NF-κB” to of TLR4 and its downstream target NF-κB.

Results:

Overall, the results are appropriately described and visualized. F values need to be added in all sections and I have a few editorial suggestions.

In Figure 1B, please indicate the significance levels above the 20 mg Tab bar.

Section 2.1: Please provide the F and p values for the ANOVA tests of the results shown in Figures 1B and 1C.

Section 2.1: Please change “Tab could alleviate the…” to Tab alleviated the…

Section 2.2: Please provide appropriate citations at the end of the first sentence.

Section 2.2: Please provide the F values for the ANOVA tests of the results shown in Figures 2A and 2C. Figure 2C does not show up in the pdf manuscript document but I assume it will be present in the resubmission.

Section 2.2: Please change “…while Tab could inhibit the expression…” to …while Tab significantly inhibited the expression…

Section 2.3: Please change “While Tab treatment could significantly inhibit the…” to Tab treatment significantly inhibited the…

Section 2.3: Please provide the F values for the ANOVA tests used.

Section 2.4: Please provide the F values for the ANOVA tests used.

Section 2.5: Please change all sentences that use …could… as suggested in other sections above.

Discussion:

Overall the discussion is short and appropriate although the second paragraph needs to be clarified.

First paragraph: Please change the second sentence to: Of 1881 drugs developed from 1981-2019, 49.2% were directly developed from natural products or natural product derivatives [16].

First paragraph, last sentence: Please provide appropriate citations.

Second paragraph, first sentence: Please change to: Microglia play an important role in the regulation of the brain microenvironment and are usually found in an inactivated state [17].

Second paragraph, second sentence: Please provide appropriate citations.

The second paragraph is confusing as it is unclear whether the authors describe their own results or are referring to published work. This is unclear because the authors include citations inconsistently in this section and throughout the manuscript.

The third paragraph is clear and appropriate.

Methods:

Overall the methods are appropriate. The authors should cite previous work that informed the methods they used in each section as indicated below. I also have some editorial suggestions.

Section 4.1: Please refer to previously published work to justify the way the BV2 microglial cells were cultured.

Section 4.2: At the end of the first sentence, please provide appropriate citations to justify the methods used.

Section 4.2: Please provide the rationale for the 4 h treatment with Tab before LPS was added for 24 h.

Section 4.3: At the end of the first sentence, please provide appropriate citations to justify the methods used.

Section 4.4: At the end of the first sentence and at the end of the last sentence, please provide appropriate citations to justify the methods used.

Section 4.5: Please provide appropriate citations throughout this section to justify the methods used.

Section 4.6: Please provide appropriate citations throughout this section to justify the methods used.

Section 4.7: The second sentence should be rewritten as it is not grammatically correct. Also, it should end with -80C.

Section 4.7: Please provide appropriate citations throughout this section to justify the methods used.

Section 4.8: First sentence: Please change to: Data are presented as…

Section 4.8: Third sentence: Please change to: All the data…

Round 2

Reviewer 2 Report

Dear Author,

After carefully revision, revised manuscript can be proceed further for publication.